# Cultural adaptation of a UK evidence-based problem-solving intervention to support Polish prisoners at risk of suicidal behaviour: a cross-sectional survey using an Ecological Validity Model

Amanda E Perry [ID],[1] Maja Zawadzka,[2] Piotr Lapinski,[3] Keeley Moore,[4] Jaroslaw Rychlik,[2] Beata Nowak[2]

¹Department of Health Sciences, University of York, York, UK
²The Academy of Justice, Warsaw, Poland
³Department of Occupational Therapy, University of Physical Education in Warsaw, Faculty of Rehabilitation, Warsaw, Poland
⁴HMPPS, Ministry of Justice, Oakham, UK

**Correspondence to**
Dr Amanda E Perry;
amanda.perry@york.ac.uk

## ABSTRACT

**Objective** To complete a cultural adaptation of a UK evidence-based problem-solving intervention to support Polish prisoners at risk of suicidal behaviour.

**Design** A cross-sectional survey participatory design using an Ecological Validity Model.

**Setting** The study was a collaboration between: the Academy of Justice, in Warsaw, the University of Lodz, two Polish prisons (ZK Raciborz and ZK Klodzko) and the University of York (UK).

**Methods** The adaptation process included an examination of the use of language, metaphors and content (ie, culturally appropriate and syntonic language), the changing of case study scenarios (relevance and acceptability) and maintenance of the theoretical underpinning of the problem-solving model (intervention comprehensibility and completeness). Four stages used: (1) a targeted demonstration for Polish prison staff, (2) a wider audit of the skills with Polish prison staff and students, (3) forward and back-translation of the adapted package, and (4) two iterative consultations with participants from stages (1) and (2) and prison officers from two Polish prisons.

**Participants** Self-selecting volunteer participants included: targeted prison staff (n=10), prison staff from the wider Polish penitentiary system (n=39), students from the University of Lodz (n=28) and prison officers from two Polish prisons (n=12).

**Main outcomes and measures** Acceptability and feasibility of the training package, reported in a series of knowledge user surveys.

**Results** The recognised benefits of using the skills within the training package included: enhancing communication, reflective development, collaborative working, changing behaviour, empowering decision-making, relevance to crisis management situations and use of open-ended questions. The skills were endorsed to be used as part of future penitentiary training for prison officers in Poland.

**Conclusions** The skills had widespread appeal for use across the Polish penitentiary system. The materials were deemed relevant while adhering to the comprehensibility of the intervention. Further evaluation of the intervention

## STRENGTHS AND LIMITATIONS OF THIS STUDY

⇒ Cultural adaptations of prison interventions are rare.
⇒ An Ecological Validity Model was used to adapt a UK problem-solving intervention for use within Polish prisons.
⇒ The study used small volunteer self-selecting groups of prison staff, students who were training to be prison officers and prison officers in two Polish prisons.
⇒ Future engagement with prisoners is key to understanding more about the acceptability and feasibility of the intervention delivery.
⇒ A randomised controlled trial is needed to measure the effectiveness of the adapted problem-solving intervention.

should be explored using a randomised controlled trial design.

## INTRODUCTION

The mental health of people incarcerated in prison is recognised as an international public health concern.[1 2] Prisoners experience disproportionate levels of mental health, isolation, boredom, self-harm and anti-social violent behaviour when compared with the general population.[2–5] Additionally, prisoners are more than five times likely to suffer from a common mental health disorder (such as depression) and once diagnosed are more likely to reoffend,[6 7] accompanied by the rising demand for healthcare often outstripping the available resources[8] and continued reports from UK and European prisons of rising incidents of violent assaults, suicidal and self-harm behaviour.[9]

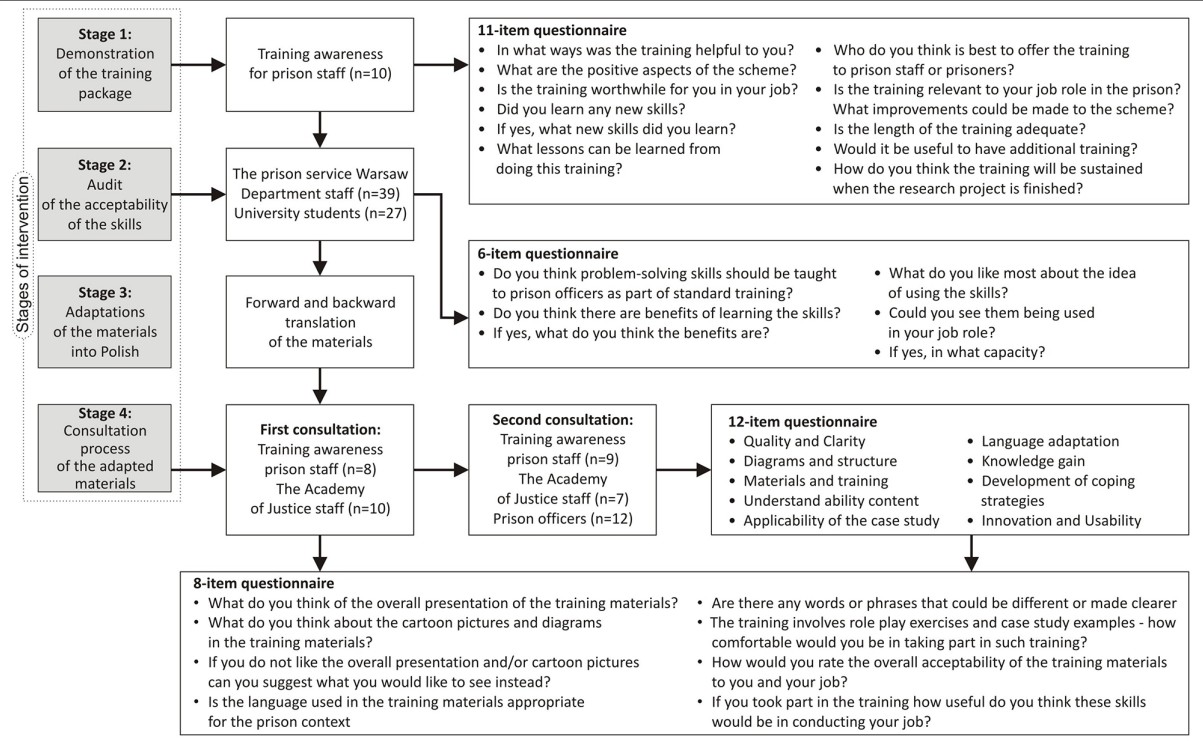

**Figure 1** Overview of study process.

International evidence on the effectiveness of interventions for suicidal people in custody (eg, multicomponent programmes, group-based programmes and peer-support interventions) shows many gaps in knowledge and limited use of high-quality research designs such as randomised controlled trials.[10] Reviews of risk factors relating to suicide and self-harm behaviour suggest that interventions should encourage social structures, develop positive relationships and provide purposeful activity by encouraging interventions that use a holistic approach to supporting the mental health of prisoners.[3 11] Despite the international recognition of the disproportionate rates of suicide and self-harm behaviour in prisons, there is little evidence to demonstrate the sharing of best practice. Often, interventions devised from community populations are not relevant without further adaptations to the prison environment.[12–15] This is particularly relevant given that poor adherence (ie, the degree to which a participant follows the recommendations of an intervention) and rates of attrition are known primary barriers to achieving optimal treatment outcomes for mental health worldwide.[16 17] Interpretation of these data is, in part, also about understanding some of the cultural differences between countries in practices and policy.

For example, Poland and the UK have similar prison populations, and official statistics of the Polish prison system show rates of suicide, self-harm and aggressive behaviour (https://sw.gov.pl/strona/Statystyka). Although rates of suicide and self-harm behaviour in Poland are relatively low (between 2015 and 2019 representing 2.53–3.38 per 10 000 prisoners),[18] cultural differences between Poland and the UK reflect different policies for the recording of self-harm behaviour. In Poland, in 2013, changes to the legal provision of recording self-harm were restricted to the inclusion of only those incidents where cause to harm lasted more than 7 days. Current self-harm rates recorded in Poland are therefore likely to be significantly less than the actual number of incidents. In addition, stigma associated with reporting mental health problems in Poland has been noted as a barrier to prisoners raising concerns about mental ill health.[19]

For many years, both the UK prison service and the Polish penitentiary service have used cognitive–behavioural programmes to address offending behaviour.[20–22] One element of such programmes includes the use of social problem-solving. In community studies, meta-analytical reviews of problem-solving skills show beneficial impact of reducing symptoms of suicidal ideation (Mean Difference −1.58, 95% CI −1.58 to −0.44), outcomes of repetition of self-harm at 4 months (OR 0.65, 95% CI 0.36 to 1.16) and final follow-up (OR 0.76, 95% CI 0.55 to 1.05).[23] Theorists[24] note that individuals with a characteristic set of negative thoughts and feelings about problems tend to blame themselves for problems and doubt their own ability to solve problems effectively. Avoidant and impulsive responses produce problem-solving and thus risk reinforcing negative beliefs and feelings that impact on mental health and well-being.[25] Use of problem-solving skills is important for many who display symptoms of depression, self-harm or violent behaviour because they often cite a problem as the main cause of their behaviour.[26] In addition, the intervention itself is relatively brief and can be delivered by anyone who is

trained to use the skills. As such, the intervention has the potential to be used by many in custody.

Prior work in the UK sought to first adapt a community-based seven-step problem-solving model producing a co-produced manualised problem-solving training package (consisting of a workbook, digital materials and handbook) for prison staff and prisoners.[27] The resulting package comprised of five 3-hour group training sessions targeting the following skills: supporting alternative styles of thinking; self-control; use of social perspective taking and consideration of consequences arising from problem-solving deficits. Key findings from the research evaluation demonstrated a promising reduction in repeat self-harm behaviour.[23] However, problem-solving skills in Poland are not used with those who present with self-harm behaviour or concerns of mental health and well-being. The study therefore aimed to adapt the UK problem-solving model using the Ecological Validity Model (EVM) process with four stages: first, a demonstration of the training materials to prison staff. This included an evaluation of the feasibility (whether something should be done, should we proceed with it and if so how) and acceptability (how well an intervention has been received by the target population). Second, a wider audit of the skills to gather feedback from the Polish prison community, followed by adaptation of the current training materials using culturally specific examples, use of forward and back-translation, and fourth, two rounds of consultations to identify further amendments to the materials.

## METHODS
### Overall study collaboration
The study was conducted using a collaboration between: the Academy of Justice, the University of Lodz, two Polish prisons (ZK Raciborz and ZK Klodzko) in Poland and the University of York (UK). Together, the two prisons hold a maximum of 1474 adult male prisoners. Each prison hosts a range of different functions for short-term prisoners and those serving life imprisonment, including a therapeutic wing for prisoners with diagnoses of mental health problems. The Academy of Justice convened the group of prison staff, identified participants for the wider audit and liaised with prison staff at each prison site. One member of the research team (MZ) held a joint position with the University of Lodz. MZ (a cognitive–behavioural therapy psychotherapist in Poland with knowledge of the scheme) was trained by the principal investigator (PI) (AEP) in the use of the intervention. The Forensic Mental Health Group and PI (AEP) (https://www.york.ac.uk/healthsciences/research/mental-health/ourresearch/forensic-mental-health/) from the University of York provided the original materials and was the project lead.

### The ecological validity process
An adapted EVM[28] included an examination of three different elements: use 'language' and 'metaphors',

'content' reflecting relevance and acceptability, and 'concept' to maintain the underpinning theoretical problem-solving model to retain the intervention comprehensibility and completeness. Language is often cited as the carrier of the culture because it presumes ease of delivery, greater familiarity with cultural knowledge and expressed emotions. As a result, culturally sensitive language may be instrumental in ensuring that the intervention is well received. Case study scenarios (originally co-produced with people in custody in the UK) were changed to reflect relevant 'content'. They were adapted to provide cultural information about values, customs and traditions that reflect the familiarity of the Polish prison system and the culture of people in Poland (figure 1).

### Knowledge user surveys
Three separate surveys were devised for stages 1, 2 and 4 of the process. In November 2019, survey one addressed the acceptability of the training and was evaluated using a brief (11-item) self-report questionnaire. The questionnaire was aimed at identifying whether the skills were acceptable and feasible to Polish prison officers. The questionnaire elicited a series of yes/no responses and semistructured qualitative comments. During January and February 2020, survey two (containing six items) was used to gather information on the wider use of problem-solving skills with Polish criminal justice staff. Between May 2020 and August 2021, surveys three (containing 8 items) and four (containing 12 items) were used to produce the final iterations of the training package.

### Procedure
#### Stage one: demonstration of the training package for Polish prison staff
The problem-solving demonstration sessions were delivered by KM and AEP in the Academy of Justice during a 3-day period in November 2019. Ten prison staff members were invited by the Director of the Academy of Justice. Apart from having a willingness to participate, staff had to meet two further criteria: (1) to have good knowledge of English and (2) to work in a position where the skills could be used in the future. The sessions included a demonstration of the problem-solving skills, sharing of the UK case study scenario examples and a series of role-play exercises to support the adaptation of the materials (see online supplemental materials). At the end of the training session, staff were asked to provide feedback using survey one.

#### Stage two: audit of the problem-solving skills with staff working in the wider Polish prison community
During January and February 2020, the audit was used to gain a wider perspective from a larger population of prison staff and students who were training to become a prison officer. MZ individually self-selected and approached staff and students to take part in the study. MZ demonstrated the problem-solving skills using a brief 90 min session using a 5 min animation and delivery of

the problem-solving skills. Staff and students were showed copies of the training workbooks before completion of survey two.

## Stage three: adaptation of the materials into Polish

Feedback from stages one and two informed the translation of the training documents. The translation from English into Polish was completed by MZ. Back-translation was completed by an independent translation service in the UK. In addition, the translation was checked against the original text by the project PI (AEP). Ratification of the text was completed with further discussion with colleagues in Poland to ensure the wording and language were comparable and culturally appropriate.[29]

## Stage four: consultation process of the adapted materials

Staff already engaged in stages one and two were asked to take part in two rounds of consultation. Additionally, in round two, 12 prison officers were recruited from two Polish prisons to review the training package. During the approach by MZ, 10 prison officers declined to take part due to the 'pressures of daily duties'. The findings from the consultation process were categorised into changes relating to the EVM including language, metaphor and content (relevance and acceptability), and content (intervention comprehensibility and completeness).

## Data analysis

All survey results were anonymised, and data were entered into a series of different Excel spreadsheets. IBM SPSS Statistics (V.26) and Statistica StatSOFT V.13 were used to explore the quantitative responses generating mean scores and descriptive statistical information. The distribution of the data was examined to identify use of either parametric or non-parametric testing at $p < 0.05$. The qualitative comments were summarised at each stage in the study and input into Excel software. Inductive coding was used to synthesise the evidence and then generate themes.[30] The Consolidated criteria for Reporting Qualitative research checklist (http://www.equator-network.org/reporting-guidelines/coreq/) was completed to ensure transparency of the methodology. Coding was completed by MZ and AEP generating themes. Qualitative comments within each theme were additionally 'counted' to provide a numerical frequency, which enabled a direct comparison of comments within each theme. The responses from stage two were divided into those reported either by prison staff or students in training.

## Patient and public involvement

The patient and public involvement group were involved in the design of the adapted workbooks through a series of consultations. The feedback from this engagement determined the final product.

## RESULTS
### Stage one: demonstration of the training package for Polish prison staff
#### Characteristics of the training sample

Prison staff (n=10) were aged between 29 and 39 years of age (mean age 34.7 years, SD=3.25). All identified as white European and spoke Polish as their first language. Most of the group participants were female (7 of 10 (70%)); one participant considered themselves disabled. Participants had worked in the prison service for an average of 103.4 months (SD=39.8, min=51 months, max=178 months), and within the current prison for a mean number of 65.7 months (SD=35.5, min=6, max=111). Prison staff had different roles, including psychologists, penitentiary departmental managers (providing substantive supervision over psychologists and educators) and specialists from the Central Board of the Prison Service. All staff reported dealing with an incident of self-harm or attempted suicide within an average of 16 months prior (SD=15.8, min=1 month, max=36 months). Of those reported incidents, 1 of 10 (10%) reported self-poisoning, 3 of 10 (30%) reported general self-injury, and 4 of 10 (40%) reported attempted suicide. Incidence of violence was reported by 7 of 10 (70%), and 3 of 10 (30%) reported an incident that could have been avoided. Most prison staff (7 of 10 (70%)) encountered prisoners with mental health problems on a daily or weekly basis (2 of 10 (20%)).

#### Acceptability of the training package and use of the skills

All participants thought that the training was useful and most learnt new skills (table 1). Five main themes included: an opportunity to learn new skills ('new knowledge' and a 'different approach is always useful'), providing useful communication skills ('learning how to speak to prisoners'), developing reflection and use of perspective taking ('to view it through the prisoner's perspective'), use of a collaborative approach ('working towards a joint search for a solution, not imposing one's views'), and use of open-ended questions to progress behaviour change ('work on changes in thinking is important'). Most staff felt that training could be delivered by prisoners, prison staff, psychologists, educators and practitioners within the prison site. Half of the group wanted the training to be longer, and nearly all thought that an additional booster session would be useful.

### Stage two: audit of the problem-solving skills with staff working in the wider Polish prison community
#### Characteristics of the audit sample

Sixty-seven volunteer participants took part in the wider audit including: 39 of 67 (58%) prison staff and 28 of 67 (42%) students. Prison staff included 30 of 39 (77%) men with ages ranging from 20 to 50 years of age. Just under one-third (12 of 39 (31%)) worked as educators, 3 of 39 (7%) were in prison management, 15 of 39 (38%) worked in the security department, 10 of 39 (25%) were shift commanders, 1 of 39 (2%) worked in the finance

**Table 1** Results following demonstration of the training package for Polish prison staff

| Questionnaire | n/10 (% yes) | Qualitative comments |
|---|---|---|
| In what ways was the training helpful to you? | Not Applicable | Development of a new perspective<br>A new technique/tool kit/alternative approach<br>Exchange of ideas between the UK & Poland |
| What are the positive aspects of the scheme? | Not Applicable | Straightforward, simple, easy to understand<br>Problems solved by prisoners<br>The development of coping skills<br>Group work<br>Providing a structure to help change |
| Is the training applicable for your job role? | 10/10 (100) | NA |
| Did you learn any new skills? | 7/10 (70) | NA |
| If yes, what new skills did you learn? | Not Applicable | Working from the prisoner's perspective<br>How to speak to prisoners<br>Development of problem-solving skills |
| Who do you think is best to offer the training to prison staff or prisoners? | Not Applicable | Prisoners, prison staff, psychologists, educators, practitioners within the prison site |
| Is the training relevant to your job role in the prison? | 9/10 (90) | NA |
| What improvements could be made to the scheme? | Not Applicable | Development of more practice exercises<br>Use of additional case study examples<br>Development of the materials |
| Is the length of the training adequate? | 5/10 (50) | The training should be longer |
| Would it be useful to have a booster session? | 9/10 (90) | NA |
| How will the training be sustained? | Not Applicable | In everyday use<br>Training of prisoners on an individual basis<br>Training as part of prisoner officer rules |

department, 2 of 39 (5%) worked in the quartermaster department, 3 of 39 (7.8%) worked in the therapeutic department, and 2 of 39 (5%) were prison inspectors. Students were aged between 20 and 30 years of age and just under one-third (9 of 28 (32%)) were male. All students were enrolled on a master's degree in National Security and upon completion would join the Polish prison service.

### Overall results from the audit

Table 2 shows the overall findings from the audit. Both prison staff and students thought that the problem-solving skills should be taught as part of standard training. The perceived benefits were wide ranging. Prison staff were more likely than students to recognise use of the skills as an opportunity to change behaviour, to empower others to increase the pace of decision-making (28 of 67 (42%)), develop knowledge (30 of 67 (45%)) and apply the skills in instances of crisis management (18 of 67 (27%)). Students reported use of the skills in general problem-solving (22 of 67 (33%)). Staff were more likely than students (37% vs 11%) to use the skills for self-development and knowledge. Students were more likely than staff (35% vs 15%) to use the skills to develop their own coping strategies. Over half of the prison staff (23 of 39 (58%)) and most of the students (19 of 28 (70%)) saw relevance of the skills within their job role. No significant differences were found in the opinion of students or staff in the perceived benefits of integrating the skills into the

training scheme for newly qualified prison officers ($X^2(1, N=66)=3.45$, $p>0.05$). Prison staff, when compared with students, were significantly more likely to accept that the training could be translated into skills for prisoners ($U=51$, median rank: 243; 222, $p<0.05$).

### Stage three: adaptation of the materials into Polish

Feedback from stages one and two enabled the materials to be translated into Polish using a series of changes within the structure of the EVM. These included:

### Language, metaphors, content

Words were added to the workbooks (eg, 'to be' or 'to have') to ensure that the context of the prison environment was more understandable in the Polish language. Words were also changed to reflect different cultural differences. In addition, some English words were nonexistent in Polish and in these cases, similar words were found and replaced to ensure that the comprehension of the materials remained the same.

### Relevance and acceptability

One male case study scenario was reviewed as part of the training package (online supplemental materials). Changes to the case study scenario ensured its applicability to the Polish penitentiary system. Such changes included the consideration of the daily routine of prisoners and the signposting of available resources in Poland (which were inevitably different to those in the

**Table 2** Audit of the acceptability of the problem-solving skills with a wider Polish prison community

| Questionnaire items | Prison officer (PO) (N% yes) | Student (S) (N% yes) | PO and S qualitative themes | |
|---|---|---|---|---|
| Do you think problem-solving skills should be taught to POs as part of standard training? | 38/39 (97) | 23/27 (85) | Not Applicable | Not Applicable |
| Do you think there are benefits from learning the skills? | 34/39 (87) | 24/27 (88) | Not Applicable | Not Applicable |
| If yes, what are the benefits? | Not Applicable | Not Applicable | Changing behaviour (PO=9/33 27% vs S=2/24 8%) Crisis planning & prevention (PO=2/33 6% vs S=3/24 12%) Developing coping strategies (PO=2/33 6% vs S=3/24 12%) Developing personal skills (PO=1/33 3% vs S=1/24 4%) Developing social networks (PO=4/33 12% vs S=1/24 4%) Generating solutions (PO=1/33 3% vs S 1/24 4%) Helping someone (PO=0/33 0% vs 5/24 20%) General problem-solving (PO=2/33 6% vs S=8/24 33%) Empowerment to make decisions at pace (PO=14/33 42% vs 2/24 8%) | |
| What do you like most about the idea of using the skills? | Not Applicable | Not Applicable | Ability to compromise (PO=3/32 9% vs S=0/17 0%) Knowledge & self-development (PO=12/32 37% vs S=2/17 11%) Developing coping strategies (PO=5/32 15% vs S=6/17 35%) Empowerment (PO=6/32 18% vs S=5/17 29%) Tailored to individual need (PO=1/32 3% vs S=0/17 0%) Improved communication (PO=1/32 3% vs S=2/17 11%) Accessible to help others (PO=4/32 12% vs S=2/17 11%) | |
| Could you see them being used in your job role? | 23/39 (58) | 19/27 (70) | Not Applicable | Not Applicable |
| If yes, in what capacity? | Not Applicable | Not Applicable | Solving problems (PO=1/22 4% vs S=3/16 18%) Helping others (PO=0/22 0% vs S=3/16 18%) Applicability in life experiences (PO=0/22 0% vs S=2/16 12%) Developing empathy (PO=0/22 0% vs S=2/16 12%) Crisis management (PO=6/22 27% vs S=4/16 25%) Increased knowledge (PO=10/22 45% vs S=1/16 6%) Empowerment (PO=3/22 13% vs S=1/16 6%) Social cultural activities (PO=2/22 4% vs S=0/16 0%) | |

UK). Options generated from step four of the problem-solving model that did not exist within the Polish context were changed to reflect options that would be acceptable to Polish prisoners.

### Intervention comprehensibility and completeness
The workbook titles were changed to reflect the comments made during stages 1 and 2. The theoretical underpinning of the intervention was adhered to throughout the structure of the workbooks to ensure that the intervention comprehensibility was maintained. British names of resources were changed to Polish resources to reflect the completeness of the intervention.

### Stage four: consultation process of the adapted materials
#### Characteristics of the sample: first and second consultations
Two rounds of consultation were included in round one: (1) prison staff from stage one (8 of 10 (80%)), and a (2) wider group of professionals (10 of 39 (25%)) from stage two. Round one results showed that most people (83%) liked or really liked the pictures and diagrams, 89%

reported both that the language reflected the context of the prison environment and the skills were perceived as useful within daily work. One-hundred per cent of participants reported that they would like to attend training. Round two included those participants who took part in round one plus a group of serving prisoner officers (n=12) from two Polish prisons who had not previously seen the materials (table 3).

### Language, metaphors, content
The combined results from the qualitative consultations in rounds one and two showed that overall, the language used was appropriate for the prison context ('I rate the materials at a level appropriate to the subjects' language; 'I think it is understandable and also simple'). The second consultation identified some useful further iteration on use of appropriate language to ('unify terminology' and reflected that 'some of the words were changed to reflect words more familiar to the Polish culure'). Based on this feedback, a final edit of the materials was completed.

**Table 3** Consultation process of adapted materials in round two

| Second consultation (n=28) | Definitely not (yes) | Rather not (yes) | No strong preference | Rather (yes) | Definitely (yes) |
|---|---|---|---|---|---|
| Quality | 0 | 1 (4%) | 2 (7%) | 15 (53%) | 10 (36%) |
| Clarity | 0 | 0 | 5 (18%) | 12 (43%) | 11 (39%) |
| Diagrams | 0 | 1 (4%) | 2 (7%) | 13 (46%) | 13 (46%) |
| Structure | 0 | 2 (7%) | 2 (7%) | 15 53%) | 8 (29%) |
| Materials | 0 | 0 | 4 (14%) | 17 (60%) | 6 (21%) |
| Training modules | 0 | 2 (7%) | 2 (7%) | 14 (50%) | 9 (32%) |
| Understanding clarity and content | 0 | 1 (4%) | 5 (18%) | 14 (50%) | 7 (25%) |
| Acceptability of the case study | 1 (4%) | 1 (4%) | 2 (7%) | 15 53%) | 8 (29%) |
| Language adaptation | 1 (4%) | 1 (4%) | 2 (7%) | 14 (50%) | 10 (36%) |
| Possibility of self-development | 1 (4%) | 1 (4%) | 5 (18%) | 18 (64%) | 3 (11%) |
| Development of coping strategies | 0 | 2 (7%) | 3 (11%) | 14 (50%) | 8 (29%) |
| Level of innovation: use within the system | 0 | 1 (4%) | 3 (11%) | 14 (50%) | 9 (32%) |

## Relevance and acceptability

The workbooks contained diagrams and pictorial representations through cartoon-like characters. These were used in the original UK workbooks to aid literacy levels.[31 32] In Poland, most prison staff thought these were acceptable with wider benefit to other client groups recognised ('the presented training materials are interesting and their great value is universalism'; 'they can be implemented with different groups of people'). The general opinions about the overall presentation of the workbooks were good ('diagram and pictures facilitate understanding of the presented content').

## DISCUSSION

To our knowledge, this study is the first to demonstrate use of the EVM to develop the cultural adaptation of an evidence-based UK problem-solving skills training package for prison staff and prisoners at risk of self-harm, mental health and well-being in Poland. Cultural adaptation procedures are key in the acceptability and recognised feasibility of using any intervention.[12] Iterative input from Polish prison staff was vital in generating a training package with the potential to support good adherence. The guiding principle of cultural adaptation is to work closely with stakeholders during early developmental stages to ensure that they can be sustained.[32 33]

The results demonstrated acceptance of the skills regardless of staff or student group, indicating that the use of the skills had potential widespread appeal across the Polish penitentiary system. Similar findings were found in the prior UK studies with high levels of acceptability and relevance reported by prison staff and prisoners.[24] As expected, staff, in comparison with students, could envisage greater application of the skills within the prison context. This may well reflect the experience and knowledge of prison staff who were already working within the Polish prisons. Prisons are challenging settings to implement the development of new skills, whereby the nature of the environment presents complex issues of trust, confidentiality and fairness with arguably many logistical and practical barriers to implementation.[34] Failure to generate interventions without high levels of acceptability or feasibility is a barrier to widespread use of interventions that may ultimately bring benefit to reductions in self-harm, and promote mental health and well-being. Interventions that engage staff from across the prison sites help consider the integration and development of sustainable models to prevent self-harm behaviour and promote mental health and well-being going forward.

## Limitations

The study has several limitations. First, the volunteer self-selecting nature of the sampling is unlikely to reflect the representative nature of all staff in the Polish prison service. Arguably, the nature of the sampling within this evaluation may impact on the generalisability of the training package. Extensions of this work would require feedback from a diverse range of prisoners (including females) and use of novel participatory strategies. Although feedback from staff reported that the skills would be relevant in circumstances of 'crisis management', further evaluation of the training package should gather views on the perceived impact on the management of suicidal and self-harm behaviour.

## Implications and future research

Our adaptation processes resulted in good levels of acceptability and feasibility, with positive feedback from prison staff and students. Recommendations refer to the necessity of using problem-solving skills as a part of standard training, by integrating skills such as role-play to support communication skills. Further development of the workbook and training package needs to consider further adaptations that will help to ensure that the materials are widely accessible to prisoners with different

protected characteristics. The inclusion of further narrative case studies for females in prison would require additional consideration.

## CONCLUSION

The process of cultural adaptation enables researchers to efficiently tailor evidence-informed interventions for diverse populations while retaining the core components that underpin efficacy. The EVM provides a useful framework for adapting interventions. Use of this method produced an acceptable training package for the Polish prison service. Further evaluation of the effectiveness of the problem-solving training skills needs to be explored using a randomised controlled trial design.

**Acknowledgements** With many thanks to the UK HMPPS Intervention services, to all the prison staff who contributed to the project including the prisons of ZK Racibórz and ZK Kłodzko, and to the Central Board of Prisons in Poland who gave permission for this research.

**Collaborators** The International Adaptation of Problem-Solving Skills (IAPSS): Kamila Nowak, Anna Szwedowska, Irmina Chrzanowska, Krzysztof Kasjan, Agnieszka Nowogrodzka, Izabela Stankiewicz, Przemysław Worobiej, Anna Kondraciuk, Bartosz Jaroszewski, Sebastian Lizyńczyk.

**Contributors** AEP and MZ devised the study with the initial support from PL. AEP and KM delivered the training in stage one. JR conducted the analysis and BN supported Polish colleagues in the written work for the publication. All elements of the project were overseen by AEP. All authors worked on the draft of the manuscript. AEP is responsible for the guarantor.

**Funding** This study was funded through the Wellcome Trust, Centre for Future Health, Rapid Response Fund, and University of York, UK.

**Competing interests** None declared.

**Patient and public involvement** Patients and/or the public were involved in the design, or conduct, or reporting, or dissemination plans of this research. Refer to the Methods section for further details.

**Patient consent for publication** Obtained.

**Ethics approval** This study involves human participants and ethical approval for the study was granted by the Department of Health Sciences at the University of York in the UK and reviewed by a RODO Data Protection Specialist in the Academy of Justice, in Warsaw, Poland. No reference number provided. Permission was granted from each prison in taking part in the study. All study participants received full informed consent.

**Provenance and peer review** Not commissioned; externally peer reviewed.

**Data availability statement** Data are available upon reasonable request.

**ORCID iD**
Amanda E Perry http://orcid.org/0000-0002-0279-1884

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
