## [Reviewer comments · BMJ Open]

ARTICLE DETAILS

TITLE (PROVISIONAL)	Cultural adaptation of a UK evidence-based problem-solving intervention to support Polish prisoners at risk of suicidal behaviour: A cross-sectional survey using an Ecological Validity Model.
AUTHORS	Perry, Amanda; Zawadzka, Maja; Lapinski, Piotr; Moore, Keeley; Rychlik, Jaroslaw; Nowak, Beata

VERSION 1 – REVIEW

REVIEWER	Dirceu Mabunda Universidade Federal de São Paulo
REVIEW RETURNED	31-Dec-2022

GENERAL COMMENTS	The theme of the study is very interesting. I would like to suggest you to include the materials of the interventions that were discussed in the trainings as annex. Regarding the qualitative analysis it's not clear how did you carried out it, which theory did you used- important to describe step by step from the coding process until themes development. I think that you should add some quotes to clarify your findings. I suggest also to add a checklist of Qualitative process. In your discussion you stated that: "Poor adherence to interventions is a known primary barrier to achieving optimal treatment outcomes for mental health worldwide" it's not clear the connection of this with you findings- I suggest to put this in the introduction and emphasize the need to do cultural adaptation. Is important to clearly discuss in which findings your study differ from what we already know and what is the similarities with other studies in the field. I would like also to suggest you to increase more references in the discussion section.
---

REVIEWER	Louis Favril Ghent University
REVIEW RETURNED	11-Jan-2023

GENERAL COMMENTS	This is an original study on the cultural adaptation of a self-harm intervention in custody. The manuscript would be a welcome addition to the literature - however, some points require clarification. * In the introduction on page 5, the emphasis seems on PST as an intervention for depression and hopelessness - which is not the same as risk of self-harm/suicide. What's the primary focus and to what extent does PST differ for these distinct outcomes (mental health vs. suicide)? Psychosocial interventions addressing mental health symptoms may or may not have downstream effects on reducing risk of suicide. * More detail regarding the content, delivery, duration, sessions, etc.
--

	of PST (as originally implemented in UK prisons) would be welcome so readers unfamiliar with the approach have a clearer idea of its key components. Also, was this originally developed for suicide risk or mental health more generally? * “Whilst different practices are used globally...” (page 5) - which ones? Please discuss psychosocial alternatives to PST in custodial settings. * Some more background on self-harm and suicidal behaviour in prisons (eg, prevalence and risk factors) is needed in the introduction - and how PST may address this. * More info should be provided on the qualitative analysis - “thematic framework” (page 9) is very vague; I’d like the authors to elaborate on this. * On page 9 it is stated that patients were involved - please describe who and how. * Most readers (including myself) are likely not familiar with the Polish prison system, their characteristics, culture etc. Please discuss these - as they are necessary to interpret the study findings. * Are students valid respondents to evaluate the acceptability of an intervention which they may not be very familiar with? What’s the advantage of including students (some of whom were trained to join the army - not prison officers)? This comes out of the blue - the authors should motivate their inclusion and describe their background knowledge of PST before participating. * Stage 3 and 4: the changes largely seem to concern wording and comprehensibility, less so a “cultural” adaptation (as emphasised in the introduction). The latter point is, in my opinion, not highlighted sufficiently. Which were the key cultural changes made? How does this relate to the Polish (vs. UK) context? Now, as a reader, I don’t see which cultural adaptations were made and which were necessary.
--	--

REVIEWER	Bryce E. Stoliker University of Saskatchewan, Centre for Forensic Behavioural Science and Justice Studies
REVIEW RETURNED	03-Apr-2023

GENERAL COMMENTS	Overall, the manuscript presents interesting insight into the cultural expansion of a training program/package aimed at addressing self-harm and suicidal behaviour among people in custody. While I am supportive of this study, I have identified some areas of concern/comments for the authors to address (listed below). Notably, a major overarching limitation centres on the relevance, acceptability, and feasibility of the content itself. In this case, the authors appeared to have placed greater emphasis on investigating the structure of the training package (which is agreeable); however, they did not appear to directly investigate participants’ views on problem-solving skills training as it relates to addressing incidents of self-harm and suicidal behaviour. General Comments: - Consider proofreading to address errors concerning grammar, syntax, sentence structure, etc. There are many areas throughout
--

with errors and difficulty with readability.

ABSTRACT

- Pg. 5, line 11: error in "...at risk of suicidal and/or self-harm." Likely intended to be "suicidal behaviour."

- Pg. 5, line 26-27: In terms of main outcomes, what exactly is meant by "acceptability and feasibility of the training package"? For instance, is it related to cost effectiveness, performance of the program, cross-cultural relevance, utility of the knowledge/skills gained by those intended to implement the programming, etc.?

- Pg. 5, line 29-30: The Results section of the abstract is used to highlight methodological processes as opposed to identifying the key findings. The authors should revise this section of the structured abstract to highlight key findings, not methodology.

- Currently, the abstract clearly highlights that the study is about cultural adaptation of the UK problem-solving skills intervention to fit the context of Polish correctional institutions; however, it is not clear what the major evaluative components are in this regard. In other words, what exactly is of interest in terms of adapting the program and evaluating its success in its adaptation (e.g., what are the key factors that are being investigated)?

INTRODUCTION

- Currently, the introductory section of the manuscript somewhat highlights the major focus of the study; however, it is not fully clear how this study fits within the context of research on suicide and self-harm and, most importantly, how it follows from existing research on this topic by Dr. Amanda Perry (I am basing this latter sentiment on the assumption that the current study is an extension, to some extent, of the previous work on problem-solving skills training). The following should be taken into consideration to provide a clearer backdrop for the current study:

1) The authors should prioritize a focus on current trends with respect to the nature of suicide and self-harm in correctional institutions to introduce the study, drawing upon the latest studies in this area (not currently cited), for example:

<https://doi.org/10.1016/j.cpr.2022.102190>

[https://doi.org/10.1016/S2468-2667\(20\)30233-4](https://doi.org/10.1016/S2468-2667(20)30233-4)

<https://doi.org/10.1111/sltb.12638>

<https://doi.org/10.1192/j.eurpsy.2020.101>

2) Related to the above point, the authors should incorporate (if possible) the latest statistics on suicide, suicidal behaviour, and/or self-harm within the Polish correctional system (and UK as a comparison) to provide a background on the current extent of the issue.

3) The paragraph that discusses the importance of cross-national sharing of best practices (pg. 6, line 18-28) and the paragraph that discusses the importance of problem-solving skills training as a component of cognitive behavioural therapy (p. 6, line 52-55 to pg. 7, line 8-19) are each relevant to the purpose of the current study. However, a few things should be addressed. First, these paragraphs should be integrated. Second, the narrative should emphasize the aim to address suicidal behaviour and self-harm. Third, these paragraphs have neglected to at least briefly highlight key findings

from previous research on the UK problem-solving skills training (i.e., Perry et al. 2019; Perry et al., 2021; etc.) in terms of addressing suicidal behaviour and self-harm. Related to the latter point, this should emphasize the known benefits of the programming at this point, especially in terms of what may be expected when adapting for prison organizations in other jurisdictions.

4) The paragraph on pg. 6, line 30-49 is relevant given the focus on problem-solving therapy, however, it could be condensed, and statistics could be removed. Related to point 3, it could be added into a condensed/integrated paragraph that emphasizes the aim of the current study.

METHODS

- Pg. 7, line 27-33: It's not entirely clear who was involved in collaboration versus who makes up each distinct group of participants. The parties included in collaboration, participation, etc. could be made clearer (providing clear distinction between participants vs. collaborators).

- Pg. 8, line 46-49: What were the different job roles of the prison staff (n = 10) who work in the Polish prison service included in stage one? Does "prison staff" include prison officers? If not, why are prison officers not included in this stage?

-Pg. 8, line 44-54: Could the authors provide examples of the problem-solving skills associated with the training package/program (or cite to studies that provide examples)? Could the authors provide an example of what role-play exercises consist of? Essentially, a description of the major aspects of the training package/program would be helpful (or citations to studies that provide fuller detail).

- Pg. 9, line 20-25: What led to the use of a purposive volunteer sample for Polish prison staff from the Warsaw Department (n = 39) and university students at the University of Lodz (n = 27)? That is, why exactly were these individuals selected and, most importantly, what steps were taken with respect to recruitment?

- Related to the above point, what parameters surrounded the recruitment of prison staff, prisoner officers, and other study participants? Further, what is the rationale for including specific participant groups in each of the stages (aside from stage two)? For instance, is there a reason why only "prison staff" were included in stage one and why the new group of prison officers were included in stage four only?

- Related to Procedure (Pg. 8-10), it does not appear that all collaborators/participants are described. For instance, who exactly was involved in stage three and who was the "new group of staff" involved in stage four? Where do the "Academy of Justice" collaborators fit in? etc.

- Pg. 10, line 19-21: The first two sentences of the "Data Analysis" subsection are not clear.

- Pg. 10, line 29-31: Were responses in other stages divided according to participant group (and not just stage two)? For example, in stage four with the introduction of several groups?

- Pg. 10, line 33-34: It is not clear what is meant by the use of

	“statistical assumptions to identify use of parametric or non-parametric testing” as the use of parametric vs. non-parametric hypothesis tests is determined by variable structure/distribution. If referring to something else, please clarify. - Pg. 10, line 47-51: How exactly were “patients” and the “public” involved in the consultation process, how many were involved, and how were they recruited/selected? What defines “patient” and “public”? - Notably, the Methods section provides little indication as to how this training package/program focuses specifically on suicidal behaviour and self-harm and, most importantly, the evaluation questions appear to primarily assess the structure of the package/program and how it operates as opposed to its content. Thus, there are no direct connections between the training package/program and the aim of preventing suicidal behaviour and self-harm. Were participants asked to reflect in any way on problem-solving skills training in relation to suicidal behaviour and self-harm? If so, why is this not included? If not, what limitations emerge in terms of better understanding the utility of the training package/program with respect to addressing suicidal behaviour and self-harm among people in custody? It is understood that the main focus of this study is on cultural adaptation of the training package/program, however, not investigating participants’ perspectives on the major aim of the program presents a major limitation. In sum, it was interesting to see that the evaluation questions did not specifically assess participants’ views on the training package/program in terms of applying it to the management of suicidal behaviour and/or self-harm. RESULTS - Pg. 11, line 26-30: Were data collected on the cumulative number of incidents each prison staff managed with respect to self-harm and/or suicide attempts (as opposed to only a yes/no for dealing with a recent incident)? This would apply to other stages as well, if this data were collected. - Pg. 11, line 16-40: As above, what were the occupational characteristics (i.e., job role) of prison staff? Related to this point, are these prison staff often dealing with people in custody directly or do they hold managerial roles and largely manage prison officers instead? The authors should also clarify that the prison staff are from the two male prisons, so as not to confuse with the “wider Polish prison community.” Similarly, the authors should clarify that the “wider Polish prison community” includes those outside of the two male prisons. Related to the latter two points, I’m basing this on an assumption. It’s not entirely clear whether this is the case, hence the benefit of further clarification. - Pg. 11, line 46-55 to Pg. 12, line 7-20: As above, it would be helpful if a description of the training package/program was provided (or a citation to these details). For instance, what exactly does it entail, how long is it, is it one time only, etc. (especially considering participants remarked on some of these as noted on pg. 11, line 46-50). - Pg. 12, line 18-20: This line is redundant to the first line in the paragraph, “half the group wanted the training to be longer...”
--	--

	- Related to some points above, the authors should use consistent and clear language when discussing each of the distinct participant groups (i.e., in terms of job roles). For instance, on pg. 14 the authors describe the audit group as both prison staff and prison officers. From a North American perspective, these terms may be interchangeable to some extent, but “prison staff” often refers to anybody working in a prison and “prison officer” would be considered a “correctional officer.” Perhaps the terms differ in a Polish and UK perspective but, in any case, consistent language would be best to ensure occupational role is not confused. - Pg. 16, line 15+: Who was involved in adapting the materials into Polish and what role did they play in each aspect? Pg. 17, line 31: With respect to Table 3, why are findings from only the second round of consultations presented and not the first? What was the specific purpose of each of the two rounds of consultation? Relatedly, what exactly is Table 3 describing in terms of the reported data? This is not explained. DISCUSSION - Pg. 19, line 11-18: While the study assessed the acceptability and feasibility of the problem-solving skills training package/program, there does not appear to be analyses on potential for adherence. Relatedly, what is meant by the authors in terms of adherence? Is this referring to the long-term adoption of the training and use of skills by prison staff/officers? Does it refer to the appropriate adoption of training/skills (i.e., fidelity)? The authors should provide further clarification or remove this concluding point. - Pg. 18-19: The discussion surrounding the acceptance and feasibility of the training package/program within the context of the current study is good. However, this discussion could be enhanced by drawing comparisons from earlier studies on the problem-solving skills training package/program. For instance, how do current findings compare to those in the UK? - Pg. 20, line 17-31: It is agreed that the sample, and sampling strategy, presents a major limitation (especially with respect to excluding people in custody as a key group with lived experience that could provide insight). Another major limitation that is not mentioned is the fact that the current study did not directly assess perceptions on problem-solving skills training with respect to addressing incidents of self-harm and suicidal behaviour. That is, what were participants’ views on the main purpose of the training package/program.
--	---

VERSION 1 – AUTHOR RESPONSE

Reviewer: 1

Dirceu Mabunda, Universidade Federal de São Paulo

Comments to the Author:

The theme of the study is very interesting. I would like to suggest you to include the materials of the interventions that were discussed in the trainings as annex.

We have added some elements of the intervention materials as supplementary materials, see note on page 7 under stage one sub-heading and in the supplementary materials and in the additional uploaded file.

Regarding the qualitative analysis it's not clear how did you carried out it, which theory did you used- important to describe step by step from the coding process until themes development.

We used the Braun and Clarke (2013) approach to reviewing the data and applied a thematic analysis using inductive coding to create themes. We have added some more text under the 'data analysis' section to comply also with the report of the qualitative methodology using the COREQ checklist.

I think that you should add some quotes to clarify your findings.

Quotes form part of our feedback in stage 3 of the project – see integration of these quotes on pages 9 under subheading 'Acceptability of the training package and use of the skills' and pages 12 under the sub-heading 'language metaphors and content'.

I suggest also to add a checklist of Qualitative process.

We have added the COREQ checklist (see comment above)

In your discussion you stated that: "Poor adherence to interventions is a known primary barrier to achieving optimal treatment outcomes for mental health worldwide" it's not clear the connection of this with you findings- I suggest to put this in the introduction and emphasize the need to do cultural adaptation.

We have moved this text to the introduction – see page 4.

This is particularly relevant given that poor adherence (i.e., the degree to which a participant follows the recommendations of an intervention) and rates of attrition are known primary barrier to achieving optimal treatment outcomes for mental health worldwide^{16,17}.

Is important to clearly discuss in which findings your study differ from what we already know and what is the similarities with other studies in the field. I would like also to suggest you to increase more references in the discussion section.

We have reviewed further studies (see also Reviewer 3 comments about use of references in the introduction section) and added some more comments – see the additional list of references at the end of this document.

Reviewer: 2

Dr. Louis Favril, Ghent University

Comments to the Author:

This is an original study on the cultural adaptation of a self-harm intervention in custody. The manuscript would be a welcome addition to the literature - however, some points require clarification.

* In the introduction on page 5, the emphasis seems on PST as an intervention for depression and hopelessness - which is not the same as risk of self-harm/suicide. What's the primary focus and to

what extent does PST differ for these distinct outcomes (mental health vs. suicide)? Psychosocial interventions addressing mental health symptoms may or may not have downstream effects on reducing risk of suicide.

Thank you for your comment – we have added some more information about the direct evidence which supports the impact of Problem Solving skills on risk of self-harm suicide; the ordering of the outcomes from the meta-analysis includes the outcomes of suicidal ideation, and repetition of self-harm at 4 months now appear first in this list. We have altered the text on pages 4 to read:

In the UK HMPPS and the Polish penitentiary service, cognitive behavioural programmes have been used for many years to address offending behaviour^{20,26-27}. One element of such programmes includes the use of social problem-solving. In the community, meta-analytical reviews of problem-solving skills show beneficial impact of reducing symptoms of suicidal ideation (MD -1.58, CI 95% -1.58 to -0.44), outcomes of repetition of self-harm at 4 months (OR 0.65, CI 95% 0.36 to 1.16) and final follow-up (OR 0.76, CI 95% 0.55 to 1.05)²³. Theorists²⁴ note that individuals with a characteristic set of negative thoughts and feelings about problems tend to blame themselves for problems and doubt their own ability to solve problems effectively. Avoidant and impulsive responses results in in-effective problem-solving and thus risk reinforcing the negative beliefs and feelings and impacting on mental health and well-being²⁵. Use of problem-solving skills is important for many who display symptoms of depression, self-harm, or violent behaviour because they often report the main cause is a problem in their lives²⁶.

In the UK HMPPS a co-produced manualised problem-solving training package (consisting of a workbook, digital materials, and handbook) for use with prison staff and delivery of the skills by prisoners was developed²⁷. Comprising of five 3-hour group training sessions and using a six-step cognitive model targeting the following skills (e.g., supporting alternative styles of thinking; self-control; use of social perspective taking and consideration of consequences arising from problem-solving deficits). Key findings from research evaluating the package demonstrated a promising reduction in repeat self-harm behaviour²³. However, problem-solving skills in Poland are not used with those who present with self-harm behaviour or concerns of mental health and well-being.

* More detail regarding the content, delivery, duration, sessions, etc. of PST (as originally implemented in UK prisons) would be welcome so readers unfamiliar with the approach have a clearer idea of its key components. Also, was this originally developed for suicide risk or mental health more generally?

Thank you – see additional text in the comment above.

* “Whilst different practices are used globally...” (page 5) - which ones? Please discuss psychosocial alternatives to PST in custodial settings.

We have added the following text and new reference (see pages 4-5):

International evidence on the effectiveness of interventions for suicidal people in custody (e.g.,

multicomponent programs, group-based programmes, and peer-support interventions) show many gaps in knowledge and limited use of high-quality research designs such as randomised controlled trials,¹⁶ and reviews of risk factors relating to suicide and self-harm behaviour suggest that interventions should encourage social structures, develop positive relationships and provide purposeful activity by encouraging interventions that utilize a holistic approach to supporting the mental health of prisoners^{3,19}.

* Some more background on self-harm and suicidal behaviour in prisons (eg, prevalence and risk factors) is needed in the introduction - and how PST may address this.

See response to comments above.

* More info should be provided on the qualitative analysis - "thematic framework" (page 9) is very vague; I'd like the authors to elaborate on this.

We have added the following text under the Data Analysis sub heading see page 8:

The qualitative comments were summarized at each stage in the study, inductive coding was used to synthesise the evidence and then generate themes³⁰. The COREQ checklist (<http://www.equator-network.org/reporting-guidelines/coreq/>) were reported to ensure transparency of the methodology. Coding was completed by MZ and AP generating themes. Qualitative comments within each theme were additionally 'counted' to provide a numerical frequency which enabled a direct comparison of comments within each theme. The responses from stage 2 were divided into those reporting either by Prison Staff (PS) or students (S) in training (Table 2).

* On page 9 it is stated that patients were involved - please describe who and how.

This is an error in the text and has been removed.

* Most readers (including myself) are likely not familiar with the Polish prison system, their characteristics, culture etc. Please discuss these - as they are necessary to interpret the study findings.

We have added the following text see page 4:

*Poland and the UK have similar prison populations and official statistics of the Polish prison system show rates of suicide, self-harm and aggressive behaviour (<https://sw.gov.pl/strona/Statystyka>). Although, rates of suicide and self-harm behaviour in Poland are relatively low; (between 2015-2019 representing 2.53-3.38 per 10,000 prisoners)¹⁰, cultural differences between Poland and the UK reflect different policies for recording of self-harm behaviour. In Poland, changes to the legal provision of recording self-harm in 2013 was restricted to the inclusion of only those incidents where cause to harm lasted **more** than seven days. Current self-harm rates recorded in Poland are therefore likely to be significantly less than the actual number of incidents. In addition, stigma associated with reporting mental health problems in Poland has been reported as a barrier to prisoners raising concerns about mental ill health¹¹.*

** Are students valid respondents to evaluate the acceptability of an intervention which they may not be very familiar with? What's the advantage of including students (some of whom were trained to join the army - not prison officers)? This comes out of the blue - the authors should motivate their inclusion and describe their background knowledge of PST before participating.*

A demonstration of the skills were presented to the students using workbook materials and an animation in Polish. Students were included in the sample as an opportunity to identify whether potential new staff members could envisage benefits from using the skills as equal value to those that were working in the field already. We have added the following text to elaborate on the inclusion and knowledge of PST before participation see page 7

MZ individually self-selected and approached students and staff to take part in the study. MZ demonstrated the problem-support skills with use of a brief 90 session that included an animation and teaching of the skills. Staff and students were showed copies of the training workbooks before completion of survey two.

** Stage 3 and 4: the changes largely seem to concern wording and comprehensibility, less so a "cultural" adaptation (as emphasised in the introduction). The latter point is, in my opinion, not highlighted sufficiently. Which were the key cultural changes made? How does this relate to the Polish (vs. UK) context? Now, as a reader, I don't see which cultural adaptations were made and which were necessary.*

The main cultural changes included:

Use of different words/language to describe the same context – and words that didn't exist/phrases that would not be used in Poland were changed to provide contextual relevance. Adaptations to the case study scenario which presented a large family of siblings in the UK; to make this more representative of Polish families the size of the family were changed (see supplementary file). In addition, relevant signposting to additional resources that were used in the UK system weren't applicable for use in Poland. For example; St Giles Trust were changed to resources that were applicable to the Polish system.

Reviewer: 3

Dr. Bryce E. Stoliker, University of Saskatchewan

Comments to the Author:

Overall, the manuscript presents interesting insight into the cultural expansion of a training program/package aimed at addressing self-harm and suicidal behaviour among people in custody. While I am supportive of this study, I have identified some areas of concern/comments for the authors to address (listed below). Notably, a major overarching limitation centres on the relevance, acceptability, and feasibility of the content itself. In this case, the authors appeared to have placed greater emphasis on investigating the structure of the training package (which is agreeable); however, they did not appear to directly investigate participants' views on problem-solving skills training as it relates to addressing incidents of self-harm and suicidal behaviour.

General Comments:

- Consider proofreading to address errors concerning grammar, syntax, sentence structure, etc. There are many areas throughout with errors and difficulty with readability.

Thank you the manuscript has been proof read and many changes have been made to improve the readability of the text.

ABSTRACT

- Pg. 5, line 11: error in "...at risk of suicidal and/or self-harm." Likely intended to be "suicidal behaviour."

This text has been changed to suicidal behaviour:

- Pg. 5, line 26-27: In terms of main outcomes, what exactly is meant by "acceptability and feasibility of the training package"? For instance, is it related to cost effectiveness, performance of the program, cross-cultural relevance, utility of the knowledge/skills gained by those intended to implement the programming, etc.?

We have added the following text to the main manuscript to explore what is meant by acceptability and feasibility. The additional text and definition of these terms now appears on pages 6 under the sub-heading 'Knowledge user surveys'.

The questionnaire was aimed at identifying whether the skills were acceptable (how well an intervention has been received by the target population) and feasible (whether something should be done, should we proceed with it and if so how) to Polish prison officers.

- Pg. 5, line 29-30: The Results section of the abstract is used to highlight methodological processes as opposed to identifying the key findings. The authors should revise this section of the structured abstract to highlight key findings, not methodology.

We have removed the original text from the results section and added this into the methodology of the abstract. The results section in the abstract now reads:

Recognised benefits of using the training package included: enhancing communication, reflective development, collaborative working, changing behaviour, empowering decision making, use in crisis management situations and use of open-ended questions. The skills were endorsed to be used as part of future penitentiary training for prison officers.

- Currently, the abstract clearly highlights that the study is about cultural adaptation of the UK

problem-solving skills intervention to fit the context of Polish correctional institutions; however, it is not clear what the major evaluative components are in this regard. In other words, what exactly is of interest in terms of adapting the program and evaluating its success in its adaptation (e.g., what are the key factors that are being investigated)?

See response to this in the reviewer comments above.

INTRODUCTION

- Currently, the introductory section of the manuscript somewhat highlights the major focus of the study; however, it is not fully clear how this study fits within the context of research on suicide and self-harm and, most importantly, how it follows from existing research on this topic by Dr. Amanda Perry (I am basing this latter sentiment on the assumption that the current study is an extension, to some extent, of the previous work on problem-solving skills training). The following should be taken into consideration to provide a clearer backdrop for the current study:

1) The authors should prioritize a focus on current trends with respect to the nature of suicide and self-harm in correctional institutions to introduce the study, drawing upon the latest studies in this area (not currently cited), for example:

<https://doi.org/10.1016/j.cpr.2022.102190>

[https://doi.org/10.1016/S2468-2667\(20\)30233-4](https://doi.org/10.1016/S2468-2667(20)30233-4)

<https://doi.org/10.1111/sltb.12638>

<https://doi.org/10.1192/j.eurpsy.2020.101>

Thank you we have reviewed the links and included the following text and references see the end of this document.

2) Related to the above point, the authors should incorporate (if possible) the latest statistics on suicide, suicidal behaviour, and/or self-harm within the Polish correctional system (and UK as a comparison) to provide a background on the current extent of the issue.

We have added the following reference which speaks to some evidence of the rate of suicide in the Polish prison system (see page 4):

Lizińczyk, S. (2023). Characteristics of suicides committed in Polish prisons, 2015–2019: Charakterystyki samobójstw popełnianych w polskich więzieniach w latach 2015-2019. Archives of Criminology, (XLII/2), 313–336. <https://doi.org/10.7420/AK2020P>.

3) The paragraph that discusses the importance of cross-national sharing of best practices (pg. 6, line 18-28) and the paragraph that discusses the importance of problem-solving skills training as a component of cognitive behavioural therapy (p. 6, line 52-55 to pg. 7, line 8-19) are each relevant to the purpose of the current study. However, a few things should be addressed. First, these paragraphs should be integrated.

We have restructured the whole introduction section of the paper in line with all reviewers' comments. This prior text is now reconfigured to take into consideration other reviewer comments and has been addressed. See new revised Introduction section of the paper.

Second, the narrative should emphasize the aim to address suicidal behaviour and self-harm. Third, these paragraphs have neglected to at least briefly highlight key findings from previous research on the UK problem-solving skills training (i.e., Perry et al. 2019; Perry et al., 2021; etc.) in terms of addressing suicidal behaviour and self-harm. Related to the latter point, this should emphasize the

known benefits of the programming at this point, especially in terms of what may be expected when adapting for prison organizations in other jurisdictions.

We have added the following text see page 5:

In the UK HMPPS a co-produced manualised problem-solving training package (consisting of a workbook, digital materials, and handbook) for use with prison staff and delivery of the skills by prisoners was developed²⁷. Comprising of five 3-hour group training sessions and using a six-step cognitive model targeting the following skills (e.g., supporting alternative styles of thinking; self-control; use of social perspective taking and consideration of consequences arising from problem-solving deficits). Key findings from research evaluating the package demonstrated a promising reduction in repeat self-harm behaviour²³.

4) The paragraph on pg. 6, line 30-49 is relevant given the focus on problem-solving therapy, however, it could be condensed, and statistics could be removed. Related to point 3, it could be added into a condensed/integrated paragraph that emphasizes the aim of the current study.

This information has been reconfigured and the statistics removed.

METHODS

- Pg. 7, line 27-33: It's not entirely clear who was involved in collaboration versus who makes up each distinct group of participants. The parties included in collaboration, participation, etc. could be made clearer (providing clear distinction between participants vs. collaborators).

We have added the following text to provide clarity between the participants and collaborators see page 6:

Overall study collaboration

The study was conducted using a collaboration between: The Academy of Justice, in Warsaw, the University of Lodz, two Polish prisons (ZK Raciborz and ZK Klodzko), and the University of York (UK). The Academy of Justice convened the group of prison staff; identified participants for the wider audit; and liaised with prison staff at each prison site. One member of the research team (MZ) held a joint position with the University of Lodz. The Forensic Mental Health Group and PI (AP) from the University of York provided the original materials and was the project lead.

- Pg. 8, line 46-49: What were the different job roles of the prison staff (n = 10) who work in the Polish prison service included in stage one? Does "prison staff" include prison officers? If not, why are prison officers not included in this stage?

We have added the following text page 9:

Prison staff had different roles, including psychologists, penitentiary departmental managers (providing substantive supervision over psychologists and educators) and specialists from the Central Board of the Prison Service

-Pg. 8, line 44-54: Could the authors provide examples of the problem-solving skills associated with the training package/program (or cite to studies that provide examples)? Could the authors provide an

example of what role-play exercises consist of? Essentially, a description of the major aspects of the training package/program would be helpful (or citations to studies that provide fuller detail).

See response to prior comments and also the new supplementary evidence file.

- Pg. 9, line 20-25: What led to the use of a purposive volunteer sample for Polish prison staff from the Warsaw Department (n = 39) and university students at the University of Lodz (n = 27)? That is, why exactly were these individuals selected and, most importantly, what steps were taken with respect to recruitment?

See comments to prior reviewers comments about how samples were selected and approached at each stage of the process.

- Related to the above point, what parameters surrounded the recruitment of prison staff, prisoner officers, and other study participants? Further, what is the rationale for including specific participant groups in each of the stages (aside from stage two)? For instance, is there a reason why only “prison staff” were included in stage one and why the new group of prison officers were included in stage four only?

Prison staff in stage one were invited to attend as representatives of the penitentiary system in Poland by invitation of the Director. This group was relatively small (n=10) because it involved training in a series of workshop sessions which required group activities and role play examples.

We wanted to add in new prison staff (those who had not been involved in the development of the package) so that we had a ‘fresh perspective’ from prison officers working in two prison sites to see what further amendments might be required that differed from those that had already been involved in the process.

- Related to Procedure (Pg. 8-10), it does not appear that all collaborators/participants are described. For instance, who exactly was involved in stage three and who was the “new group of staff” involved in stage four? Where do the “Academy of Justice” collaborators fit in? etc.

We have provided some extra context for the role of the collaborators – see earlier comment from another reviewer and new text that was added to page 6 under the sub heading ‘study collaboration’.

- Pg. 10, line 19-21: The first two sentences of the “Data Analysis” subsection are not clear.

We have changed the text it now reads: see page 8 under Data Analysis sub heading

All survey results were anonymized and data were entered into a series of different Excel spreadsheets. IBM SPSS Statistics (version 26) and Statistica StatSOFT version 13 were used to explore the quantitative responses generating mean scores and descriptive statistical information.

- Pg. 10, line 29-31: Were responses in other stages divided according to participant group (and not just stage two)? For example, in stage four with the introduction of several groups?

No in stage 4 responses from the total sample (n=30) were combined and presented in one Table – see Tables 3 and 4.

- Pg. 10, line 33-34: It is not clear what is meant by the use of “statistical assumptions to identify use of parametric or non-parametric testing” as the use of parametric vs. non-parametric hypothesis tests is determined by variable structure/distribution. If referring to something else, please clarify.

We have clarified the text under the data analysis heading – see page 8:

The distribution of the data were examined to identify use of either parametric or nonparametric testing at ($p < 0.05$).

- Pg. 10, line 47-51: How exactly were “patients” and the “public” involved in the consultation process, how many were involved, and how were they recruited/selected? What defines “patient” and “public”?

This text was reported in error and has been removed.

- Notably, the Methods section provides little indication as to how this training package/program focuses specifically on suicidal behaviour and self-harm and, most importantly, the evaluation questions appear to primarily assess the structure of the package/program and how it operates as opposed to its content. Thus, there are no direct connections between the training package/program and the aim of preventing suicidal behaviour and self-harm.

Research evidence shows a connection between repeat self-harm and suicidal behaviour and meta-analytical analyses shows this to demonstrate a reduction in self-harm behaviour – see more elaboration of the connection between self-harm and suicidal behaviour and problem-solving skills now listed in the Introduction part of the paper.

The training package as you state does not address self-harm or suicidal behaviour per se but challenges what problems might underly the behaviour (i.e. what is causing/worrying the person such that they act in this manner). Most people who self-harm report ‘problems in their lives’ as being the main reason for their behaviour.

Were participants asked to reflect in any way on problem-solving skills training in relation to suicidal behaviour and self-harm? If so, why is this not included?

We were primarily interested in evaluating whether the training package could be adapted; what the adaptations were and whether what was produced was seen as feasible and acceptable to staff that would be using the information. Acceptance of the training package were an important consideration; without this input research has shown that it would be less likely to be used in practice. Participants were asked to consider use of the skills within their current job roles and many cited relevant use of the skills in situations of crisis management. We have added a further limitation of the study to encourage further consultation specifically on the use of the package for suicidal and self-harm behaviour to acknowledge this concern.

If not, what limitations emerge in terms of better understanding the utility of the training package/program with respect to addressing suicidal behaviour and self-harm among people in custody?

To some extent in Poland this is not known as it was not explored in our study. The next step would be to undertake a primary high-quality study to use the training skills to see whether indeed they did make any difference in addressing suicidal behaviour and self-harm for people in custody in Poland. In this UK we did find this to be the case (see Perry et al 2021).

It is understood that the main focus of this study is on cultural adaptation of the training package/program, however, not investigating participants’ perspectives on the major aim of the

program presents a major limitation. In sum, it was interesting to see that the evaluation questions did not specifically assess participants' views on the training package/program in terms of applying it to the management of suicidal behaviour and/or self-harm.

Please see our comment above relating to why this was not the case. We have also acknowledged this as an additional limitation under the limitation sub-heading of the paper adding the following text on page 14:

Although feedback from staff reported that the skills would be relevant in circumstances of 'crisis management'; further evaluation of the impact of the training package should also gather views on the perceived impact on the management of suicidal and self-harm behaviour.

RESULTS

- Pg. 11, line 26-30: Were data collected on the cumulative number of incidents each prison staff managed with respect to self-harm and/or suicide attempts (as opposed to only a yes/no for dealing with a recent incident)? This would apply to other stages as well, if this data were collected.

No data were only collected in a demographic questionnaire that reported on the categorical variable 'yes/no' for dealing with a recent incident.

- Pg. 11, line 16-40: As above, what were the occupational characteristics (i.e., job role) of prison staff? Related to this point, are these prison staff often dealing with people in custody directly or do they hold managerial roles and largely manage prison officers instead? The authors should also clarify that the prison staff are from the two male prisons, so as not to confuse with the "wider Polish prison community." Similarly, the authors should clarify that the "wider Polish prison community" includes those outside of the two male prisons. Related to the latter two points, I'm basing this on an assumption. It's not entirely clear whether this is the case, hence the benefit of further clarification.

We have clarified the text

The job roles of the prison staff included a mixture of prison officers, psychologists and managerial roles. The prison staff referred to in stage 1 were invited from across the Polish correctional service – so they were an unrelated group to the 'wider Polish prison community which included the two male prisons'

- Pg. 11, line 46-55 to Pg. 12, line 7-20: As above, it would be helpful if a description of the training package/program was provided (or a citation to these details). For instance, what exactly does it entail, how long is it, is it one time only, etc. (especially considering participants remarked on some of these as noted on pg. 11, line 46-50).

We have added the following text to the introduction section see prior comments to other reviewers above and in addition provided a supplementary file of information (see also prior responses to similar comments above).

- Pg. 12, line 18-20: This line is redundant to the first line in the paragraph, "half the group wanted the training to be longer..."

Thank you we have removed this text from this section.

- Related to some points above, the authors should use consistent and clear language when discussing each of the distinct participant groups (i.e., in terms of job roles). For instance, on pg. 14 the authors describe the audit group as both prison staff and prison officers. From a North American

perspective, these terms may be interchangeable to some extent, but “prison staff” often refers to anybody working in a prison and “prison officer” would be considered a “correctional officer.” Perhaps the terms differ in a Polish and UK perspective but, in any case, consistent language would be best to ensure occupational role is not confused.

Thank you – we have changed this reference to ‘prison officer’ in the audit section to ‘prison staff’. We have also changed this terminology in Table 2 which presents the findings of the audit from this group of staff members. In Table 2 this staff group are now referred to as PS – Prison Staff instead of PO – Prison Officers.

- Pg. 16, line 15+: Who was involved in adapting the materials into Polish and what role did they play in each aspect?

We have added the following text – page 8:

Stage three: Adaptation of the materials into Polish

Feedback from stages one and two informed the translation of the training documents. The translation from English into Polish was completed by MZ. Back translation was completed independently by a translation service in the UK. In addition, the translation was checked against the original text by the project PI (AP). Ratification of the text was completed with further discussion with colleagues in Poland to ensure the wording and language was comparable and culturally appropriate ²⁹.

Pg. 17, line 31: With respect to Table 3, why are findings from only the second round of consultations presented and not the first? What was the specific purpose of each of the two rounds of consultation? Relatedly, what exactly is Table 3 describing in terms of the reported data? This is not explained.

We have added the results from round one in a narrative within the text and added some more information on Table 3.

DISCUSSION

- Pg. 19, line 11-18: While the study assessed the acceptability and feasibility of the problem-solving skills training package/program, there does not appear to be analyses on potential for adherence. Relatedly, what is meant by the authors in terms of adherence? Is this referring to the long-term adoption of the training and use of skills by prison staff/officers? Does it refer to the appropriate adoption of training/skills (i.e., fidelity)? The authors should provide further clarification or remove this concluding point.

Adherence in this paper refers to people remaining in the intervention (so prisoners who might receive the intervention staying in the group until the intervention is complete). In prisons adherence to interventions is generally poor. Other research has quoted that ensuring that interventions are viewed as acceptable and feasible by those that deliver and receive training is more likely to retain people in receiving the full complement of the intervention. We have removed this comment from the discussion and added above the information into the Introduction. See newly revised Introduction section.

- Pg. 18-19: The discussion surrounding the acceptance and feasibility of the training package/program within the context of the current study is good. However, this discussion could be enhanced by drawing comparisons from earlier studies on the problem-solving skills training package/program. For instance, how do current findings compare to those in the UK?

We have drawn in some further comparisons from the UK adaptation process – see information presented on page 13

The results demonstrated acceptance of the skills regardless of staff or student group, indicating that the use of the skills had potential widespread appeal across the Polish penitentiary system. Similar findings to use of the package in the UK were reported by prison staff and prisoners²⁴

- Pg. 20, line 17-31: It is agreed that the sample, and sampling strategy, presents a major limitation (especially with respect to excluding people in custody as a key group with lived experience that could provide insight). Another major limitation that is not mentioned is the fact that the current study did not directly assess perceptions on problem-solving skills training with respect to addressing incidents of self-harm and suicidal behaviour. That is, what were participants' views on the main purpose of the training package/program.

We agree and have added this to our 'limitations section' and see other comments addressing this above.

We have removed some references and added others to respond the reviewer comments. This list below shows the additional references within the manuscript:

Favril, L., Shaw, J., Fazel, S. (2022). Prevalence and risk factors for suicide attempts in prison, *Clinical Psychology Review*, Volume 97, <https://doi.org/10.1016/j.cpr.2022.102190>.

Favril, L., O'Connor, R., Hawton, K., & Vander Laenen, F. (2020). Factors associated with the transition from suicidal ideation to suicide attempt in prison. *European Psychiatry*, 63(1), E101. doi:10.1192/j.eurpsy.2020.101

Ministry of Justice 2022 Safety in Custody Statistics, England and Wales: Deaths in Prison Custody to December 2017 Retrieved from National Statistics: Ministry of Justice, 2022: <https://www.gov.uk/collections/safety-in-custody-statistics>. Accessed April 2023.

Vos, T., Lim, S. S., Abbafati, C., Abbasi, M., & Abbasi-Kangevan, M. e. a. (2020). Global burden of 369 diseases and injuries in 204 countries and territories, 1990-2019: a systematic review analysis of the Global Burden of Disease Study 2019. *The Lancet*, 396(10258: 1204-1222: doi), 1204-1222: doi.org/1210.1016/SO1140-6736 (1220) 30925-30929.

Lizińczyk, S. (2023). Characteristics of suicides committed in Polish prisons, 2015–2019: Charakterystyki samobójstw popełnianych w polskich więzieniach w latach 2015-2019. *Archives of Criminology*, (XLII/2), 313–336. <https://doi.org/10.7420/AK2020P>

- Przybyliński S., Marczak M. (2008). Self-mutilation among female and male prisoners. In: W. Ambrozik, H. Machel, P. Stępnia (eds.) The mission of the Prison Service and its tasks in relation to the current penal policy and social expectations. IV Polish Penitentiary Congress (p. 455) Poznań-Gdańsk-Warsaw: UAM, UG, Pedagogium WSPR and CZSW.
- Faregh, N., Lencucha, R., Ventevogel, P., Dubale, B. W., & Kirmayer, L. J. (2019). Considering culture, context and community in mhGAP implementation and training: challenges and recommendations from the field. *Int J Ment Health Syst* 13:58. doi: 10.1186/s13033-019-0312-9.
- Carter, A. Butler, A. Willoughby, M. Janca, E., Kinner, S.A., Southalan, L., Fazel, S. Borschman, R. Interventions to reduce suicidal thoughts and behaviours among people in contact with the criminal justice system: A global systematic review. *eClinicalMedicine* 2022;44: 101266 Published online 14 January 2022 <https://doi.org/10.1016/j.eclinm.2021.101266>
- Perry, A. E., (2020). Self-harm in prisons: what do we know and how can we move forwards? www.thelancet.com/psychiatry Vol 7 August 2020, 649-650. [https://doi.org/10.1016/S2215-0366\(20\)30298-4](https://doi.org/10.1016/S2215-0366(20)30298-4).
- Reingle Gonzalez, J. M., & Connell, N. M. (2014). Mental health of prisoners: identifying barriers to mental health treatment and medication continuity. *American journal of public health*, 104(12), 2328-2333. <https://doi.org/10.2105/AJPH.2014.302043>.
- Nagpal, T.S., Mottola, M.F., Barakat, R., Prapavessis, H. (2021). Adherence is a key factor for interpreting the results of exercise interventions, *Physiotherapy*, Volume 113, Pages 8-11, <https://doi.org/10.1016/j.physio.2021.05.010>.
- Gobbett MJ, & Sellen, J.L. (2014). An evaluation of the HM prison service "thinking skills programme" using psychometric assessments. *Int J Offender Therapy Comp Criminol.*, Apr;58(4):454-73. doi: 10.1177/0306624X12472485. Epub 2013 Feb 6. PMID: 23390065.
- Perry A. E., Waterman MG, House A, et al (2018) Problem-solving training: assessing the feasibility and acceptability of delivering and evaluating a problem-solving training model for front-line prison staff and prisoners who self-harm *BMJ Open* 2019;9:e026095. doi: 10.1136/bmjopen-2018-026095.
- D'Zurilla, T. J., Chang, E. C., Nottingham, E. J. IV, & Faccini, L. (1998). Social problem-solving deficits and hopelessness, depression, and suicidal risk in college students and psychiatric inpatients. *Journal of Clinical Psychology*, 54(8), 1091–1107. [https://doi.org/10.1002/\(SICI\)1097-4679\(199812\)54:8<1091::AID-JCLP9>3.0.CO;2-J](https://doi.org/10.1002/(SICI)1097-4679(199812)54:8<1091::AID-JCLP9>3.0.CO;2-J).
- D'Zurilla, T. J., & Goldfried, M. R. (1971). Problem solving and behavior modification. *Journal of Abnormal Psychology*, 78(1), 107–126. <https://doi.org/10.1037/h0031360>.

Linehan, M. M., Camper, P., Chiles, J. A., Strosahl, K., & Shearin, E. (1987). Interpersonal Problem-Solving and Parasuicide. *Cognitive Therapy and Research*, 11(1), 1-12. doi:Doi 10.1007/Bf01183128

Perry, A. E., Waterman, M., Dale, V., Moore, K., & House, A. (2021). The effect of a problem support intervention on self-harm and violence in prison: an interrupted time series analysis using routinely collected prison data. *EClinical Medicine- The Lancet*, 32 (1007021), 1-11
<https://doi.org/10.1016/j.eclinm.2020.100702>.

VERSION 2 – REVIEW

REVIEWER	Dirceu Mabunda Universidade Federal de São Paulo
REVIEW RETURNED	26-May-2023

GENERAL COMMENTS	The authors addressed the questions raised.
---

REVIEWER	Bryce E. Stoliker University of Saskatchewan, Centre for Forensic Behavioural Science and Justice Studies
REVIEW RETURNED	19-May-2023

GENERAL COMMENTS	I believe the authors have largely addressed the questions/comments/concerns raised by reviewers from the first review. My only recommendation is to again review the manuscript for clarity in writing, grammatical errors, syntax, etc. as there are many throughout. Also, following from the comments of Reviewer 2, it would be beneficial to include some details on how the Polish prison system is structured, including things like incarceration rate, number of people in custody in the two prisons included in the study, etc.
---